# Green Manure Planting Incentive Measures of Local Authorities and Farmers' Perceptions of the Utilization of Rotation Fallow for Sustainable Agriculture in Guangxi, China

**Leonard Ntakirutimana [1] , Fuduo Li [1] , Xianlei Huang [1], Shu Wang [1] and Changbin Yin [1,2,*]**

[1] Institute of Agricultural Resources and Regional Planning, Chinese Academy of Agricultural Sciences, Beijing 100081, China; leonard.ntakirutimana@yahoo.fr (L.N.); lifuduo2010@163.com (F.L.); 82101181256@caas.cn (X.H.); wangshu@caas.cn (S.W.)

[2] Key Laboratory of Nonpoint Source Pollution Control, Ministry of Agriculture and Rural Affairs, Beijing 100081, China

* Correspondence: yinchangbin@caas.cn

**Abstract:** Planting green manure in fallow croplands in winter can bring various economic and environmental benefits, including increased food production, carbon capture and sequestration, soil retention, sandstorm prevention, water retention, and provision of habitat for biodiversity. However, the increased production cost of planting green manure reduces farmers' willingness to adopt this approach, which is unfavorable for its sustainability. This research aims to investigate the influence of instrumental variables on farmers' perceptions of sustainable agriculture practices, especially the use of rotation fallow, and tries to understand the relationship between farmers' perceptions of using rotation fallow and planting green manure under incentive measures adopted by local authorities in Guangxi Province, China. Using simultaneous equation models, the results show that subsidies and planting training were the most important drivers for restoring green manure planting in the target region. These incentive measures could be further enhanced as a priority to restore green manure planting. The study also finds that socioeconomic factors such as farmer's income, area of farmland, and labor for agricultural production have a certain influence on planting green manure planting and on farmers' perceptions of using rotation fallow as a form of sustainable agriculture practice.

**Keywords:** incentive measures; farmers' perceptions; green manure; rotation fallow; Guangxi China

## 1. Introduction

In China, the past half-century has seen remarkable growth in food production. In 2016, about 621.4 million tons of grain was produced. Food security is no longer the sole goal of agricultural development, and increased attention is now concentrated on environmental protection and sustainable agriculture development. At present, soil degradation in China has affected more than 466 million hectares. It is one of the most serious agricultural, environmental [1], and socioeconomic problems [2]. Some of the problems were caused by the overuse of chemical fertilizers. The use of chemical fertilizers went from 8.8 million tons in 1978 to 58.6 million tons in 2017, a 6.7-fold increase [3]. This excessive utilization generated many environmental problems, such as greenhouse gas emissions, water pollution, weak quality of agricultural production, and loss of biodiversity [4]. More than 40% of arable land has been affected by soil degradation (loss of soil nutrients, salinization, acidification, and weakened ecosystem services) [5]. The implications of such phenomena have been food insecurity, negative

effects on the environment, and loss of ecosystem services [1]. Therefore, it is necessary to plant green manure and use rotation fallow to improve soil quality and protect the environment.

Green manure species (e.g., *Astragalus sinicus* L., *Vicia villosa* Roth., and *Medicago sativa* L.) have different characteristics in terms of their service and function in sustainable environmental protection and agricultural economic development, such as increasing the humus in the soil and carbon sequestration [6] and improving the quality of soil fertility [7]. One study [8] pointed out the benefits of green manure planting. For these issues, after returning green manure to the soil, the crops decomposed and released their nutrients into the soil to improve its structure, which increased soil organic matter content (humus, carbon, and nitrogen), maintained the cycle of nutrients in the agricultural ecosystem, and enhanced the biomass and activities of soil microorganisms. Deng [9] compared the physical and chemical properties of soil and the yield of a succession of crops before and after planting green manure, and the results showed that the availability of potassium (K) was increased by 17.99% and organic matter by 25.45%, soil bulk density decreased by 22.55%, and pH stayed the same. If considering yield, the succeeding rice crops were increased significantly by a rate of 6.62% within two years. Planting green manure is a significant historical practice in China [10,11] and the use of fallow land in winter is a very important strategy for optimizing the agricultural cropping structure.

Planting green manure in fallow croplands in winter can bring various economic and environmental benefits, including food production, carbon capture and sequestration, soil retention, sandstorm prevention, water retention, and provision of habitat for biodiversity. However, although the government enthusiastically advocates green manure planting, implementation of the practice has been slow, and the increasing production cost of planting green manure reduces farmers' willingness to adopt this approach, which is unfavorable for its sustainability. The cost of planting green manure (*Astragalus sinicus* L.) in paddy fields in south China was 1730 RMB/hm$^2$ [12]. Currently, subsidy policies for planting green manure are still in the research and exploration stage in China, and there is no subsidy policy nationwide. Under the current policy, the increased production cost of planting green manure is fully borne by farmers, which lessens their willingness to adopt this approach. The cost of green manure planting is high in terms of economic outcomes and there are no formal or adequate incentive measures to motivate farmers to plant green manure voluntarily. Even though many researchers have recognized that there is growing environmental awareness and a public demand for improving the environment in China [13], the existing research has not paid much attention to this important topic. A systematic and rigorous study of farmers' willingness to plant green manure has been lacking. What are the key factors that influence farmers' willingness to plant green manure? How can farmers be encouraged to more sustainably plant green manure? These questions are very important, but they have not yet attracted enough attention of scholars.

Hence, we carried out research to estimate farmers' willingness to plant green manure in typical green manure production regions of China and to identify determinants underlying their willingness by using simultaneous equation models. The overall goal was to know farmers' perceptions of planting green manure as a sustainable agricultural practice and determine which factors affect their willingness. We identified indicators of efforts that could be focused on so as to sustain harmonious restoration and develop green manure planting in specific areas. This study could be helpful in encouraging farmers to engage in more sustainable planting of green manure and promoting the sustainable development of agriculture.

## 2. Literature Review

Acceptance/adoption: This concept is a psychological process that starts when a person or an operation finds something novel and finishes with adoption of the final stage [14]. The acceptance/adoption of a policy is influenced by many factors, such as the socioeconomic characteristics of the household, people's knowledge, and their awareness of the benefits or opportunities to receive benefits [15].

Green manure: Planting green manure is a traditional worldwide practice [1,14]. Conventional agricultural fertilizers are abandoned. Green manure has functions of fixing soil nitrogen, improving soil properties, and reducing the consumption of fertilizers [16], which provide economic benefits for farmers, because planting green manure provides storage for forage, ensures food security, and improves the ecological environment. In addition, depending on its potential, planting green manure affects the management of weeds, diseases, and pests in cropping fields [17]. The most important thing is that planting green manure plays an important role in developing traditional agriculture in China [11,18].

Fallow: The term "fallow" usually refers to leaving fields uncultivated for a period of time to restore soil fertility and physical characteristics. There are obvious benefits to this farming practice, because soil moisture conservation promotes nitrate accumulation and controls weeds. The efficiency of fallow depends on the cropping system, tillage methods, and different kinds of soil texture. In practice, small farmlands need the combination of rotating fallow with planting green manure to maintain soil fertility, because of the doubled population between 1949 and 1980 and the smaller per capital arable land, even down to 0.1 ha [19].

Rotation fallow: Rotation is a planned green manure practice that consists of crop succession to maintain the health of the environment and the economy of the farm [20]. This technique has the practical significance of protecting species diversity, increasing soil organic matter content, and reducing the use of chemical fertilizers and pesticides [21]. The mixed rotation of crops and green manure can control root-knot nematodes and soil-borne fungi on vegetable crops [20]. In China, planting green manure crops in fallow farmland in winter is a significant strategy to optimize the cropping structure [11]. In northern China, the ecological service value of rotating *Vicia villosa* and *Orychophragmus violaceus* with spring maize was 71,449 (*Vicia villosa*) and 69,962 (*Vicia villosa*) yuan per hectare [22]. Rotating forage and vegetable crops has benefits in terms of managing nematodes and soil-borne pathogens that are often associated with vegetable and cattle producers. In order to improve the quality of agronomic and vegetable crops in the southeastern United States, coastal bermudagrass has been used in crop rotations to reduce yield loss caused by root-knot nematode and soil-borne fungi [17]. The conventional rotation wisdom is supported by the following: (i) avoid planting the same family of crops in the field successively; (ii) plant cover crops and cash crops alternately; (iii) substitute deep-rooted crops with shallow and fine-rooted crops alternately; (iv) precede heavy feeders with nitrogen-fixing cover crops; and (v) avoid following a root crop with the same crop [20].

Government policy: Agro-environmental incentive measures encourage farmers to participate in policy implementation. Defrancesco [23] observed that in order to increase farmers' willingness to convert unoccupied rural residence property into cultivated land, communication between farmers and administrations, advertising, and transparency of rural residential property use policies should be improved. In the field of agricultural public policy, Liu [24] found that training for technical generalization of nonpoint-source pollution was helpful in reducing fertilizer consumption. The goal of agricultural policy was to improve yields by applying more irrigation and chemical fertilizers, which was called "modern conventional agriculture." The consequences of that policy were faster soil degradation and water, soil, and air pollution caused by inadequate application of chemical fertilizers; small quantities of organic matter; the letdown of wet and dry crop rotation; and especially reduced planting of green manure and legumes. Therefore, the proper rotation of crops has been neglected [19]. The National Planting Green Manure Policy could supplement existing policies on farmland supply for sustainable development such as the Land Law of the Administration of the People's Republic of China (1984–2004), with the objective of increasing the cultivated land supply to 16,000 km$^2$, the Law of Property of the People's Republic of China (2007), the Consolidation of National Land and Plan of Rehabilitation (2012), with the objective of guaranteeing cultivated land [15].

Household income based on farming: Household income is a significant factor in farmers' willingness to convert unoccupied rural residential property into cultivated land, especially growers who do not make their living or partly make their living from farming [15].

Farmer identity: Household characteristics such as age, assets, size, and education level [23,24] are important factors. Family management and land fragmentation are also major factors [24] affecting farmers' decision-making process. The knowledge of farmers who make their living entirely on farming may be affected by the policies of residential land use, awareness of the consequences of land conversion, and the size of the household [15].

Land use: Land is an important social and economic resource. Smallholder farming could be adapted to produce some crops and their comparative advantages could be emphasized, which would lead to trading products and input with other farmers. Large and medium-sized farmland can increase productivity and contribute as a supplement to family labor [25,26]. In addition, policy-makers must include farmers in planning new policy, in order to persuade reluctant farmers who do not want to adopt new technology.

## 3. Materials and Methodology

We conducted a survey on green manure planting for sustainable agricultural development in Guilin city, Guangxi Province in May 2018 (Figure 1). We chose to study this area because Guangxi Province has a long history of planting green manure (mainly *Astragalus sinicus* L.), but it has declined significantly. With the development of the chemical fertilizer industry since 1990, the planting area of green manure decreased rapidly from 6,693,104 hectares in 1991 to less than 133,104 hectares in 2012 [4]. The use of chemical fertilizers per hectare increased dramatically from 317.23 kg in 2005 to 426.57 kg in 2016. Consequently, in order to strengthen the Guangxi local government policy of restoring green manure planting initiated in 2013, we aimed to find out farmers' perceptions of the use of rotation fallow as a sustainable agricultural practice. Studying this region will be useful to understand the reasons why farmers do not plant green manure as they did before.

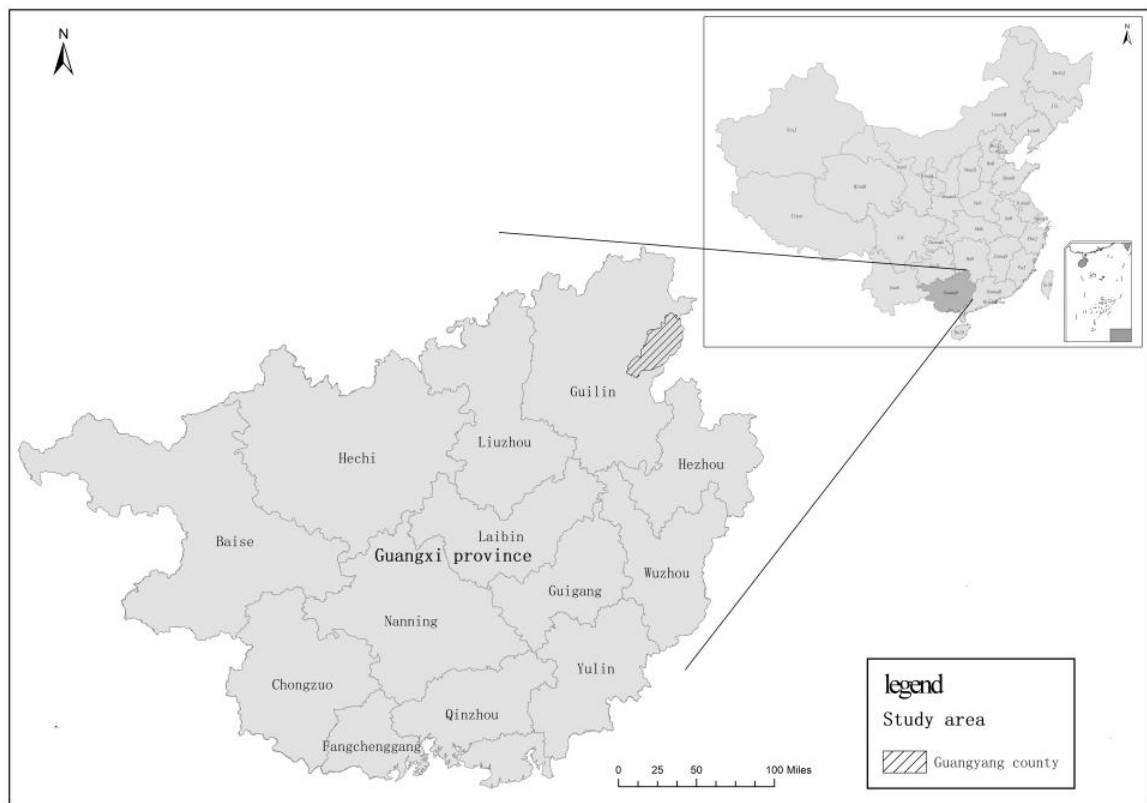

**Figure 1.** Study area.

For the survey, 336 farmers were randomly interviewed with a face-to-face questionnaire. The collected data were analyzed in Stata 14.2 (StataCorp LLC, 4905 Lakeway, Drive College Station, TX, USA). The method used to analyze farmers' perceptions of using rotation fallow was based on human behavior borrowed from neoclassical economics [27]. Alternative behavior models were based on economic theory, which predicts a problem of choice where a rational economic agent chooses the option of utility maximization [27].

The utility function of farmers' perception can be expressed as follows:

$$U_{ij} = \alpha X_i + \varepsilon_{ij} \tag{1}$$

where $\alpha$ is a constant; $X_i$ represents variables; and $\varepsilon_{ij}$ is error, where $j$ is either 1 = yes or 0 = no, and $i$ is the number of observations.

The empirical approach used in this study is based on the role of farmers' perceptions of using rotation fallow within a framework of conditional maximum likelihood [28] that considers the potential effect of factors not measured jointly affecting farmers' perception. We began with a logistic model for green manure planting (PGM). The model can be expressed with the basic probit model [29] as follows:

$$PGM_i = \beta_0 + \beta_1 Per_i + \beta_2 X_i + \varepsilon_i \tag{2}$$

where $PGM_i$ is a binary variable that takes the value 1 if the farmer participates in planting green manure and 0 otherwise; Per is a dummy variable representing the farmer's perception of using rotation fallow in order to improve the quality of natural resources and environmental protection; and $X_i$ is a vector of observed characteristics believed to affect a farmer's decision, including gender, age, education, household size, human workforce 16–65 years old, annual income, and land.

The estimation of Equation (2) did not consider the potential endogeneity factors associated with farmers' perceptions of using rotation fallow. The results are inconsistent estimates. Then, we estimated a second structural equation that estimates parameters of endogenous variables [30]. This endogeneity takes its origin through the unobserved characteristics that influence farmers' perceptions. For example, some farmers know more than others about green manure. The knowledge level could be acquired from training, membership in cooperatives, or a higher level of education than other farmers. However, to produce more reliable estimates of the influence of farmers' perceptions, the following equation system can be estimated:

$$PGM_i = \beta_0 + \beta_1 per_i + \beta_2 X_i + \mu_i \tag{3}$$

$$Per_i = \gamma_0 + \gamma_1 Z + \gamma_2 X_i + \varepsilon_i \tag{4}$$

where $Z$ is a vector of instrumental variables thought to powerfully influence farmers' perceptions and does not directly affect green manure planting. We consider the two equations as a structural model [30] and $Per_i$ and $Z$ are endogenous variables of Equations (3) and (4), respectively; $X_i$ represents exogenous variables of Equations (3) and (4). These variables include knowledge about promoting green manure planting by providing a subsidy standard, technological training or demonstration, farming experience, and receiving subsidies. An assumption about parameters in order to solve for Equations (3)–(5) is $\beta_1 \neq 0$. We assume that the error terms $\mu_i$ and $\varepsilon_i$ are uncorrelated.

We simultaneously estimated Equations (3) and (4) through methods of conditional maximum likelihood [31] with heteroscedasticity (robust standard error) [32]. The system of equations is estimated using the conditional mixed process (CMP) estimator of Stata software [28,31]. The CMP is a flexible tool to estimate systems of equations with various link functions [33]. Then, we can consistently estimate parameters of Equations (3) and (4) by using CMP [31] as follows:

$$Cmp\ (PGM_i = per_i X_i)\ (Per_i = Z\ X_i),\ ind\ (\$cmp\_probit\ \$cmp\_probit)\ technique\ (dfp)\ qui\ robust \quad (5)$$

The Davidon–Fletcher–Powell (DFP) algorithm [34] was used to resolve the difficulty in maximizing likelihood functions by eliminating the nonconcave regions and achieving convergence of the test from the estimation process. The mean and standard deviation were generated by $(\mu = E[X] = \frac{\sum_{i=1}^{n}(X)}{N})$ and $(\sigma = \sqrt[2]{\sigma^2})$ [35], respectively.

## 4. Results and Analysis

### 4.1. Demographic Characteristic of Respondents

Table 1 shows the mean and standard deviation of variables for the analysis. Growers of green manure plants ranged in age from 22 to 90 years old (average, 53.9 years), were predominantly male (61.01%), had a primary school education and below (61.12%), and had an average household size of 5.04 persons (Tables 1 and 2). Considering the agricultural labor force (16–65 years old), the majority (60.24%) of respondents had a household size of one to two members in the agricultural labor force, followed by families with a household size of three to four members (11.87%) and five to six members (0.89%) (Table 3). Most households (95.25%) did not have a family member in an agricultural enterprise or a cooperative, or in other employment such as agricultural management. Only 4.45% of respondents were represented in some agricultural organization (Table 3). Most of the respondents (44.21%) had five or six family members, followed by 37.98% of respondents with three or four family members, 8.90% with seven or eight family members, and 4.45% with one or two, or nine or more family members (Table 2), with an average of five members in the household (Table 1). In Table 2, the cumulative percentage of age shows that 81.01% of respondents were between 22 and 65 years old.

**Table 1.** Description of statistics and variable definitions (*n* = 336).

| Variable | Definition and Measurement | AV | SD |
|---|---|---|---|
| Gender | Binary variable: 1 if farmer is female, 0 if male | 0.390 | 0.488 |
| Respondent age | Age in years at the time of the survey | 53.946 | 13.820 |
| Education level | Respondent's level of completed education | 1.545 | 0.802 |
| Household size | Number of persons in the household | 5.036 | 0.546 |
| Log_income | Logarithm of total income in 2017 | 0.886 | 0.874 |
| Log_agrifield | Logarithm of the area of an agricultural field of household | 0.716 | 0.816 |
| Training | Binary variable: 1 if farmer participates in green manure planting, technological training, or demonstration; 0 otherwise | 0.116 | 0.321 |
| Log_distance | Logarithm of the distance between the home and nearest green manure demonstration | 0.596 | 1.589 |
| Farming experience | Number of years between the start of planting green manure until 2017 | 15.104 | 22.288 |
| Log_gmcost | Logarithm of the cost of green manure planting in 2017 | 0.522 | 1.291 |
| Substarec | Logarithm of standard subsidy received for green manure planting in yuan | 0.912 | 1.696 |
| Sub_fun | Binary variable: 1 if farmer believes that a standard subsidy has obvious effects on promoting green manure planting; 0 otherwise | 0.155 | 0.362 |
| PGM | Binary variable: 1 if farmer plants green manure in the agricultural field; 0 otherwise | 0.655 | 0.476 |
| per | Binary variable: 1 if farmer is willing to use rotation fallow on farmland; 0 otherwise | 0.113 | 0.317 |
| Agri_labor force | Number of household members 16–65 years old who provide labor in the agricultural field | 1.497 | 1.161 |
| Work | Binary variable: 1 if household has a member in agricultural enterprise, cooperative, or other employment such as agricultural management; 0 otherwise | 0.045 | 0.207 |
| Suppol | Binary variable: 1 if farmer prefers one of three types of subsidy (funds, seeds, mechanical services or tools); 0 otherwise | 0.342 | 0.475 |

AV, average; SD, standard deviation; PGM, green manure planting.

**Table 2.** Demographic characteristics of respondents (*n* = 336).

| Variable | Number of Respondents | Percentage (%) |
|---|---|---|
| **Sex** | | |
| Male | 205 | 61.01 |
| Female | 131 | 38.99 |
| **Age** | | |
| 20–29 | 18 | 5.34 |
| 30–39 | 39 | 11.57 |
| 40–49 | 42 | 12.46 |
| 50–59 | 90 | 26.71 |
| 60–65 | 84 | 24.93 |
| 66–69 | 19 | 5.64 |
| 70–79 | 37 | 10.98 |
| 80 and older | 8 | 2.37 |
| **Level of education** | | |
| Primary school and below | 206 | 61.12 |
| Secondary school | 90 | 26.73 |
| High school (university) | 27 | 8.01 |
| Specialist (doctoral degree) | 13 | 3.86 |
| **Household size** | | |
| 1, 2 | 15 | 4.45 |
| 3, 4 | 128 | 37.98 |
| 5, 6 | 149 | 44.21 |
| 7, 8 | 30 | 8.90 |
| 9 or more | 15 | 4.45 |

**Table 3.** Characteristics of respondents' operations (*n* = 336).

| Variables | Number of Respondents | % |
|---|---|---|
| **Perception of farmers (per)** | | |
| 1 = farmer wants to use rotation fallow in the agricultural field | 38 | 11.31 |
| 0 = otherwise | 298 | 88.69 |
| **Planting green manure (PGM)** | | |
| 1 = farmer participates in planting green manure in the agricultural field | 130 | 38.69 |
| 0 = otherwise | 206 | 61.31 |
| **Training** | | |
| 1 = farmer participates in green manure technological training or demonstration | 39 | 11.61 |
| 0 = otherwise | 297 | 88.39 |
| **Subfun** | | |
| 1 = farmer believes that standard subsidy has obvious effects on promoting green manure planting | 52 | 15.48 |
| 0 = otherwise | 284 | 84.52 |
| **Agri_labor force** | | |
| 1, 2 | 203 | 60.24 |
| 3, 4 | 40 | 11.87 |
| 5, 6 | 3 | 0.89 |
| **Work** | | |
| 1 = household has a member in an agricultural enterprise, cooperative, or other employment such as agricultural management | 15 | 4.46 |
| 0 = otherwise | 321 | 95.54 |
| **Farming experience** | | |
| 0–15 years | 224 | 66.67 |
| 16–30 years | 7 | 2.08 |
| 31–45 years | 57 | 16.96 |
| 46–60 years | 36 | 10.71 |
| 61–70 years | 12 | 3.57 |
| **Suppol** | | |
| 1 = if farmer prefers one of three types of subsidy (funds, seeds, mechanical services or tools) | 115 | 34.23 |
| 0 = otherwise | 221 | 65.77 |

### 4.2. Description of Respondents' Perceptions of Operation Variables

Among growers of green manure crops, 11.28% had an interest in using rotation fallow for several years (Tables 1 and 3). Among the respondents, 38.99% planted green manure on farmland, and 11.61% participated in green manure planting technological training or demonstration. In addition, 15.48% of respondents believed the standard subsidy for promoting green manure planting had an obvious effect. Most of the respondents (66.67%) had 0–15 years of green manure planting experience, 16.96% had 31 to 45 years of experience, and 10.71% had 46 to 60 years of experience (Table 3).

Most farmers were reluctant to plant green manure and use rotation fallow as a sustainable agriculture practice. We found that the reasons were farming experience [4] and low ecological compensation [36]. We estimated that the two models were differentiated by the number of instrumental variables. Consistency coefficients were observed in the models with fewer instrumental variables (M2). These findings show that among a big group of many elements, drivers constituted a small group of elements. In our case, three instrumental variables were included, because they had great influence on farmers' participation in green manure planting in Guangxi Province and increased their perception of using rotation fallow.

### 4.3. Farmers' Perceptions of Using Rotation Fallow and Planting Green Manure

In general, less than 50% of respondents were willing to use rotation fallow (Figure 2a) and less than 50% of farmers had a field of green manure (Figure 2b).

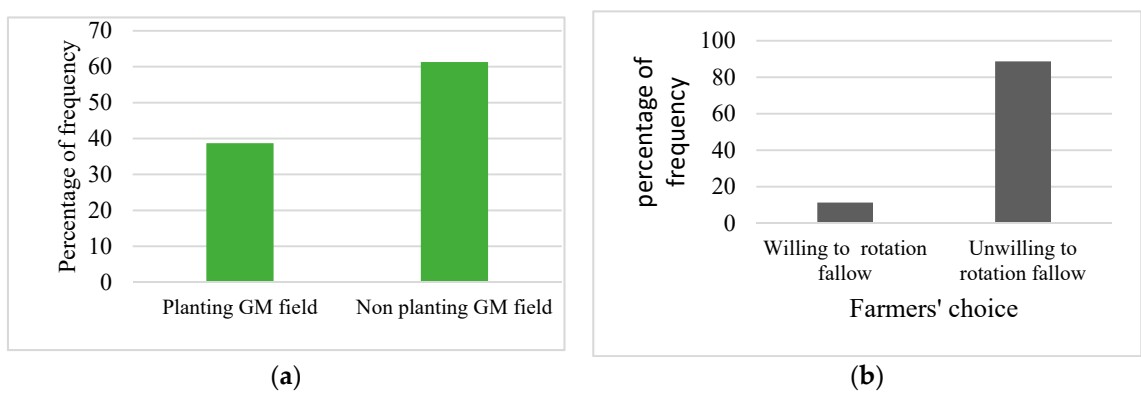

**Figure 2.** (**a**) The percentage of farmers' perceptions of rotation fallow, and (**b**) the percentage of farmers' willingness of planting green manure (GM).

An increased consistency of coefficients comes from variation of a system of instrumental variables (*SUPPOL, SUBFUN, SUBSTAREC, LOG_GMCOST, LOG_DISTANCE, TRAINING*, and *FARMING_EXPERIENCE*). Tables 4 and 5 show that the preference for a type of subsidy (funds, seeds, or mechanical services) (*SUPPOL*), farmers' participation in green manure technological training or demonstration (TRAINING), and their belief in the effects of a standard subsidy on promoting green manure planting (*SUBFUN*) were strong drivers of the program to promote green manure planting in Guangxi Province. Therefore, SUPPOL significantly influenced the farmers' perceptions. The findings also show that household members 16–65 years old who provide labor in the agricultural field (*AGRI_LABORFORCE*), the household's area of agricultural field (*LOG_AGRIFIELD*), and total income in 2017 (*LOG_INCOME*) were the exogenous factors that significantly influenced the planting of green manure.

**Table 4.** Results of conditional mixed process (Cmp) estimation and IV-postestimation tests.

| Dependents Variables | Independents Variables | M-1 | M-2 |
|---|---|---|---|
| **Planting green manure (PGM)** | | *Coeff.* | *Coeff.* |
| | per | −1.428 (5.80) ** | 1.617 (4.93) ** |
| | sex | 0.001 (0.01) | 0.080 (0.30) |
| | age | −0.011 (1.35) | −0.012 (1.23) |
| | educ | 0.007 (0.05) | −0.198 (0.89) |
| | Household_size | −0.061 (1.25) | −0.028 (0.57) |
| | work | 0.717 (1.40) | −0.205 (0.50) |
| | agri_laborforce | 0.277 (3.21) ** | 0.108 (1.54) |
| | log_agrifield | 0.841 (5.04) ** | 0.693 (3.55) ** |
| | log_income | −0.450 (4.54) ** | −0.462 (2.53) * |
| | _cons | −0.000 (0.00) | 0.277 (0.37) |
| **Farmers perceptions (per)** | *Instrumental variables* | *Coeff.* | *Coeff.* |
| | Suppol | −0.730 (3.32) ** | 1.449 (3.88) ** |
| | Subfun | 0.092 (0.15) | 0.164 (0.31) |
| | training | 0.374 (1.19) | |
| | Substarec | −0.006 (0.04) | 0.017 (0.14) |
| | Log_gmcost | 0.007 (0.08) | |
| | Log_distance | 0.142 (1.99) * | |
| | Farming experience | −0.023 (3.22) ** | |
| | *Exogenous variables* | −0.032 (0.19) | |
| | sex | −0.002 (0.22) | −0.089 (0.36) |
| | age | 0.107 (0.61) | 0.015 (1.27) |
| | educ | −0.089 (1.38) | 0.332 (1.26) |
| | Household_size | 0.468 (0.99) | −0.022 (0.45) |
| | work | 0.306 (3.26) ** | 0.409 (0.98) |
| | agri_laborforce | 0.682 (4.47) ** | 0.193 (2.79) ** |
| | log_agrifield | −0.348 (2.48) * | −0.105 (0.59) |
| | log_income | −0.963 (1.34) | 0.216 (0.78) |
| | _cons | 7.663 (3.51) ** | −3.315 (4.16) ** |
| | atanhrho_12 | 7.663 (3.51) ** | −8.509 (0.48) |
| | Rho_12 | −0.99 | −0.99 |
| | N | 336 | 336 |
| | Wald chi2 * | * (23) 192.92 | (20) 310.61 |
| | Log pseudolikelihood | −216.51 | −227.77 |
| **IV-postestimation test** **2SLS** | | | |
| Test of endogenous | Robust score chi2(1) | 0.578 (*p* = 0.45) | 21.81 (*p* = 0.00) |
| | Robust regress. F (1,325) | 0.534 (*p* = 0.47) | 40.59 (*p* = 0.00) |
| Test of first stage | R-sq. | 0.2206 Prob > F = 0.004 | 0.1772 Prob > F = 0.32 |
| Test of overidentification. | Score chi2 * | * (6) 142.022 (*p* = 0.00) | * 25.067 (*p* = 0.00) |
| **GMM** | | | |
| Test of endogenous | GMM C statistic chi2 (1) | 1.914 (*p* = 0.176) | 7.84 (*p* = 0.00) |
| Test of the first stage | R-sq. | 0.2206 Prob > F=0.004 | 0.1772 Prob > F = 0.32 |
| Test of overidentification. | Hansen's J chi2 * | * (6)142.022 (*p* = 0.00) | * (2) 25.067 (*p* = 0.00) |

Notes: *** significant at 1%, ** significant at 5%, and * significant at 10%.

**Table 5.** Average marginal effects of farmers' perceptions, factors on planting green manure, and on the utilization of rotation-fallow.

| Variables | M-1 | M-2 |
|:---:|:---:|:---:|
| Pgm | Dy/dx | Dy/dx |
| Per | −1.427 (−5.8) *** | 1.617 (4.93) *** |
| Agri_laborforce | 0.276 (3.21) *** | 0.108 (1.54) |
| Log_agrifield | 0.840 (5.04) *** | 0.693 (3.55) *** |
| Log_income | −0.450 (−4.54) *** | −0.462 (−2.53) *** |
| Per | Dy/dx | Dy/dx |
| Suppol | −0.7305 (−3.32) *** | 1.449 *** (3.880) |
| Subfun | 0.0914 (0.15) | 0.164 (0.310) |
| Training | 0.374 (1.19) | 0.017 (0.140) |
| Substarec | −0.0062 (−0.04) | |
| Log_gmcost | 0.0068 (0.08) | |
| Log_distance | 0.1418 (1.99) ** | |
| Farming_experience | −0.0229 (−3.22) *** | |

Notes: *** significant at 1%, ** significant at 5%, and * significant at 10%. Reported that average marginal effects and their heteroscedasticity robust standard errors are shown in parentheses.

## 5. Discussion

Economic incentive policies should be implemented in order to reduce the low economic impact associated with green manure planting, specifically rotation fallow, in which different practices of agriculture are integrated based on a rational use of natural resources, to increase the use of rotation fallow land as a sustainable agriculture practice. Adopting rotation fallow allows ecosystems to be maintained sustainably [20,37]. According to our survey, on average, 38.99% of farmers have participated in planting green manure and 11.28% of farmers were interested in using rotation fallow. The results show that more than 50% of farmers were reluctant to plant green manure and to use rotation fallow as a sustainable agriculture practice. This occurs in a context where most farmers are accustomed to overusing chemical fertilizers to maximize yields. This has accelerated soil degradation, resulted in failure to rotate wet and dry crops, and decreased the planting of vegetables and green manure crops [19], causing enormous consequences for the environment. Modifying traditional agricultural practices could help households by finding opportunities to increase family income and, at a high enough degree, to substantially alleviate poverty [38]. This lack of interest by farmers to adopt green manure planting and use rotation fallow contradicts the goal of the government of China to raise rural farmers' incomes and protect the environment by planting green manure and using rotation fallow as a sustainable agriculture practice.

According to our research, the first model (M-1) showed that farmers' perceptions of the use of rotation fallow has a significantly negative coefficient resulting from a combination of seven instrumental variables. The tests of goodness of fit are significant except the test of endogeneity. The model has the problem of endogeneity. Improving the consistency of estimator coefficients of the second model (M-2) required different combinations of two or three instrumental variables. These findings show that among a big group of many elements, driving factors constitute a small group of

elements. We found that three instrumental variables were essential to increase farmers' perceptions of using rotation fallow, and thus impacted significantly on their participation in green manure planting in Guangxi Province. The tests of goodness of fit are more significant.

In the second model (M-2), farmers' perceptions (per) of using rotation fallow had a significantly positive influence on planting green manure, while this factor (per) was influenced by a more significantly positive coefficient of SUPPOL and positive coefficients of SUBFUN and SUBSTAREC. Then we have the consistency coefficients of estimators in the second model (M-2). This consistency of coefficient of estimation is a result of reducing unobserved factors (IV) one by one, with the objective of finding the driving factors of farmers' perceptions of using rotation fallow as a sustainable agricultural practice. Thus, farmers' perceptions impacted more positively and significantly on their participation in green manure planting. Farmers' participation in green manure planting increased by 1.617 times with respect to their perception of using rotation fallow.

According to our study, especially regarding the driving factor (SUPPOL), farmers' perceptions will be greater than the current estimate in the future when considering their preference for various kinds of subsidy. This factor is the most suitable for understanding farmers' perceptions of maximizing green manure planting in the future. According to the driving factors, farmers' perceptions are positively affected by the standard subsidy of the local government of Guangxi based on the area planted (SUBSTAREC) and the farmers' belief that standard subsidy effects will promote green manure planting (SUBFUN). Based on the survey, 11.31% of respondents participated in training, and 34.23% had the capacity to choose one of three types of subsidy (funds, seeds, or mechanical service tools). Farming experience was more significant and had a negative impact on using rotation fallow as a sustainable agriculture practice, because 66.67% of respondents had 0 to 15 years of farming experience. Increasing training on green manure planting technology or demonstration should increase farmers' knowledge about the importance of green manure planting. Increasing the number of green manure demonstrations in different sites in Guangxi Province could be a suitable way to enhance farmer's knowledge about green manure planting technology for sustainable agricultural development. For that, a reasonable standard subsidy for the use of rotation fallow is needed to reduce the cost of green manure planting activities, because cost is one of the issues related to expenditure on labor, seeds, weed management, pest and disease control, etc.

Clearly, research evidence has shown a reluctance among farmers to participate in planting green manure and using rotation fallow. Therefore, policy-makers should focus on strengthening training sessions and raising mass awareness of the ecosystem values of green manure and added values resulting from rotation fallow practices. The use of the Internet, television, radio, and other available mass media could serve widely in advertising. Policy-makers could also take into account an ecological compensation system based on the use of rotation fallow reinforced by technical guides, technical-financial support, and technology to increase the yields of farms and the welfare of farmers.

To achieve the goal of the Chinese government, the environmental and economic incentive policy of technical-financial support can be a great help to enhance the use of rotation fallow and planting green manure and raise farmers' incomes simultaneously. The agriculture department of Guangxi has announced a series of guidelines on the development of green manure for the promotion of beautiful village construction, winter fields, and spring opportunities to develop agriculture for leisure and local tourism, because green manure planting was considered as an important part of the action plan for the increase in grain of 3 million [4]. In 2016, the planting area of green manure was 330,000 hectares and the government provided 15 million yuan in subsidies for seeds of green manure and the development of green manure demonstration regions [39].

## 6. Conclusions

The conventional agriculture adopted by the Chinese government with the use of high levels of synthetic fertilizers modified the attitudes and practices of agriculture in Guangxi, Southern China. This research aimed to assess farmers' perceptions of using rotation fallow based on incentive measures

adopted by local authorities. We carried out an analysis of the endogenous and exogenous factors that influence green manure planting and farmers' perceptions. Our methodology was adopted considering the potential endogeneity bias from unobserved factors that have an influence on farmers' perceptions but are not directly associated with green manure planting. The study used a dataset collected in May 2018 from farmers in Guangxi. Through our research, we tried to understand the relationship between farmers' perceptions of using rotation fallow and their participation in green manure planting by using conditional maximum likelihood with heteroscedasticity robust standard error. The results show that subsidizing farmers planting green manure based on a standard subsidy by the unit of sown green manure area, training on green manure planting technology or demonstration, and the preference of farmers for the kind of subsidy (funds, seeds, or mechanical services) were the most important drivers for restoring green manure planting in the target region. These incentive measures could be further enhanced as a priority to restore green manure planting. The study also finds that total income in 2017, the area of agricultural field per household, and the number of household members 16–65 years old who work in the agricultural field have a certain influence on green manure planting and on farmers' perceptions of using rotation fallow as a sustainable agriculture practice.

We observed some limitations of this research that should be noted. On the one hand, the estimated perceptions of farmers were not robust enough for application in other provinces of China because the survey was carried out in only one province. Guangxi Province is in southern China, and its circumstances are much different than other regions. Therefore, more surveys can be conducted in different provinces to strengthen the reliability and applicability of the findings. On the other hand, this study emphasized that subsidy is a key factor that influences farmers' willingness to plant green manure, but this is just the beginning. Furthermore, the expected value of subsidies should be estimated through surveys. Additionally, apart from cost, other factors such as diseases that arise by planting green manure should be taken into consideration.

**Author Contributions:** All authors were involved in preparing the manuscript. Conceptualization, L.N. and C.Y.; funding acquisition, C.Y.; methodology, L.N.; supervision, C.Y.; data curation, X.H.; writing—original draft, L.N.; writing—review and editing, X.H. and S.W., F.L. and X.H.

**Funding:** This research was funded by Major Program of National Social Science Foundation of China (Grant No. 18ZDA048), China Agriculture Research System-Green Manure (CARS-22-G-25), the National Agricultural Innovation Project of Chinese Academy of Agriculture Science for Incentive Mechanism on Agricultural Waste Resource Utilization (2016-2020), the Sino-Japan Agriculture Cooperation Project for the Food Value Chain Assessment of Green Rice (2016-2021), and the China Scholarship Council (CSC), No: CSC/2017GBJ009591.

**Acknowledgments:** We would like to thank the editor and the anonymous reviewers for their helpful comments and suggestions.

**Conflicts of Interest:** The authors declare no conflict of interest.

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
