# Peer review of "Green Manure Planting Incentive Measures of Local Authorities and Farmers’ Perceptions of the Utilization of Rotation Fallow for Sustainable Agriculture in Guangxi, China"

_sustainability, doi:10.3390/su11102723_

Round 1

Reviewer 1 Report

Comments:

 The paper “The Green Manure Planting Incentives Measures of Local Authorities and Farmers Perceptions on the Utilization of Rotation Fallow for the Sustainability of Agriculture in Guangxi, China” addresses a topic that factors influence green manure planting.

It is important for sustainable development of agriculture to planting green manure and utilizing rotation fallow. However, the systematic research on its incentive measures (include government policy and the training on the green manure planting technology and so on) is rare. In some degree, the research result can provide a theoretical guidance for developing green manure and rotation fallow.

Generally, the introduction is not well organized and the discussion need to be improved and focused on the data obtained from this study. Besides, the information provided by the manuscript lacks of substantial novelty.

This paper needs to be edited to improve the English. The language using in the current version is difficult to understand the meaning of several sentences. I sincerely suggest to the authors that, prior to resubmit their manuscript, they should carefully revise and improve it. Major editorial comments are listed below.

Suggestions:

L14-16: Rewrite.

L19: Delete “,” before “providing” and add “who”.

L31-32: Rewrite the sentence “the land used by farmers did not allow for high yields”.

L33-37: The sentence is too long and complex, please rewrite and make it easy to understand.

L37: The details of phosphorus content and the instant potassium content needn’t to describe so specific, make a short description about their change trends.

L38-39: Whether the reference “Zeng-Qiang, L. I., Jian-Hong, W., & Xian, Z.. (2017).” is the same as “Li Z. Q., (2017)”. If so, delete the sentence “The study carried out by Zeng-Qiang, L. I., Jian-Hong, W., 38 & Xian, Z.. (2017).”

L41: Change “its structure” into “soil structure”, Change “content” into “contents”.

L45: Change “potassium (K) was increased of 17.99%, the organic matter of 25.45%, soil bulk density decreased by 45 22.55 %” into “potassium (K) was increased of 17.99%, the organic matter was increased of 25.45%, soil bulk density was decreased of 22.55 %”.

L47-51: The sentence is too long and there are some grammar mistakes, please rewrite or divide the whole sentence into two or three short sentences.

L60: Add “a” before “great”.

L62: Change “use” into “used”.

L66-69:Rewrite the sentence, there are some grammar mistakes.

L70: Change “less” into “declining”.

L82: Change “look like” into “such as”.

L85: Add “in” before “worldwide”.

L87: Delete “the”.

L89: Keep verbs in the same tense. Change â€śproviding” into “provide”, change “improving” into “improve”.

L90: Delete “of”.

L97: Change “has” into “had”.

L98: Change “is” into “was”.

L99-100: “Actually, the small farmland needs the combination of rotation fallow with planting green manure.”.

L103: Change “for” into “of”.

L106: Change “content” into “contents”.

L109: Change “optimizing” into “optimize”.

L113: Change “attractive” into “attracted”.

L114-116:“In order to improve quality of agronomic and vegetable crops in the southeastern United States, coastal bermudagrass has been used in crops rotations to reduce yield loss caused by root-knot nematode and soilborne fungi.”

L135:Change “km2” into “km2”.

L142: Change “major’s” into “major”.

L147: Delete the second “and”.

L148: Rewrite the sentence.

L166: Check the format of the reference in the text.

L177-180: This paragraph repeats with L173-176.

L229: In table 1, the meaning of abbreviation “AV” and “SD” should illustrate below the table.

L257: Delete “from”.

L267: Remove extra spaces.

After L288, the line number suspended.

The format of References didn't keep consistent, please revise the References with the correct format. 

Author Response

Response to Reviewer#1’s comments: 

Thank you very much for your comments and suggestion. Changes have been made as suggested in the revised version. 

1.The abstract needs further improvements; the authors should add one or two sentences to indicate a introduction of their work 

Response: We have added two sentences to indicate an introduction of our study. “Planting green manure in fallow croplands in winter can bring various economic and environmental benefits, including increasing food production, carbon sequestration, soil retention, sandstorm prevention, water retention and provision of habitat for biodiversity. However, the increasing production cost caused by planting green manure reduces farmers’ enthusiasm to adopt this approach, which is adverse to the sustainability of planting green manure. ” (L11-L15) 

2. The global novelty of research is not highlighted in the Introduction section as well. The authors should add a paragraph and mention the novelty of this research. 

Response: We have revised the introduction section. We have added several references for some important statements to explain the importance of this study, knowledge gaps and study goals.

Firstly, we have added the sentence â€śIn China, the past half century has seen remarkable growth in food production. In 2016, about 621.4 million tons of grain was produced. Food security is no longer the sole target of agricultural development, and increased attention is now focused on environmental protection and sustainable agricultural development. At present, soil degradation in China affected more than 466 million hectares. It is one of the most serious agricultural, environmental [1] and social-economic problems [2]. Some of those problems were caused by the overuse of chemical fertilizer. The use of chemical fertilizer was passed from 8.8 million tons in 1978 to 58.6 million tons in 2017, either 6.7 times [3]. This excessive utilization generated many environmental problems such as greenhouse gases’ emissions, water pollution, and weak quality of agricultural production and loss of biodiversity [4]. More than 40% of arable land were affected by soil degradation (loss of soil nutrient, salinization, acidification and weakened of ecosystems services) [5]. The implications of that phenomena were food insecurity, negative effects on environment, loss of ecosystems services[1]. Therefore, there are a certain necessity to plant green manure and use rotation fallow to improve soil quality and to protect environment.” (L29-41) in the first paragraph of the introduction section to explain reality problems of sustainable agricultural development. 

Secondly, we have added the sentence “Although planting green manure in fallow croplands in winter can bring various economic and environmental benefits, including food production, carbon sequestration, soil retention, sandstorm prevention, water retention and provision of habitat for biodiversity. However, implementation of planting green manure practices has been slow, despite their official promotion, and the increasing production cost caused by planting green manure reduces farmers’ enthusiasm to adopt this approach, which is adverse to the sustainability of planting green manure. The cost of planting green manure (Astragalus sinicus L) in paddy fields in south China was 1730 RMB/hm2 [12]. Currently, subsidy policies of planting green manure are still in the research and exploration stage in China. There is no such a subsidy policy for planting green manure nationwide. Under the current policy, the increased production cost caused by planting green manure is fully burdened by farmers, which reduces their willingness to adopt this approach. The cost of green manure planting is high comparatively to economic outcomes and there is no formal or adequate incentive measures which could motivate farmers to planting green manure voluntarily. Even though many researchers recognized that there is a growing environmental awareness and public demand of improvement of environment in China [13]. However, the existing researches paid not much attention to this important topic. A systematic and rigorous study of intention survey of farmers’ willingness for planting green manure has been lacking. What are the key factors that influence farmers’ willingness for planting green manure? How to encourage farmers to more sustainably planting green manure? These questions are very important, but they have not yet attracted enough attentions of scholars.” (L56-L74) in the third paragraph of the introduction section to explain the problems of planting green manure.  

Finally, we have added the sentence “Hence, we carry out the research to estimate farmers’ willingness for planting green manure in typical green manure production regions of China and to identify determinants underlying farmers’ willingness by using simultaneous equation models. The overall goal is to know farmers’ perception towards planting green manure as sustainable agricultural practices and determine which factors affect their willingness for planting green manure. We identify indicators on which efforts could be concentrated so as to sustain a harmonious restoring and develop green manure planting in a specific area. This study is helpful in encouraging farmers more sustainable planting green manure and promoting sustainable development of agriculture.” (L75-L82) in the fourth paragraph of the introduction section to explain the novelty of this research. 

3. The authors should indicate how this study (results) is going to be beneficial to the policy makers in one paragraph. 

Response: We have restructured the discussion section to emphasize how to implement the policy of incentive based on our research.   

Firstly, we have added the sentence “Clearly, research evidence has shown a reluctance among farmers to participate in planting green manure and in the use of rotation fallow. Therefore, policymakers should urgently focus on strengthen  training sessions and mass awareness on the ecosystem values of green manure and on added-values resulting from rotation fallow practices. The use of Internet, television, radio and others mass media available could serve widely in advertising. Policymakers can also take into account an ecological compensation system based on the use of rotation fallow reinforced by technical guides, technical-financial support and technology to increase the yield and welfare of farmers. ” (L335-L341) in the discussion section to indicate how this study is going to be beneficial to the policy makers.  

Meanwhile, we have added the sentence “To achieve the goal of Chinese government, the environmental and economic incentive policy of technical-financial support for adopting the use of rotation fallow and planting green manure can be a great help to enhance the use of rotation fallow, planting green manure and raising farmers’ income simultaneously. The agriculture’s department of Guangxi has announced a series of guidelines on the development of green manure for the promotion of beautiful villages construction, winter fields, and spring opportunities to develop the agriculture of the leisure and the local tourism because at the same time green manure panting was considered as an important part of the action plan for the increase in grain of 3 million [4]. In 2016, the planting area of green manure is 330,000 hectare and the government provide 15 million Yuan for subsiding seeds of green manure and the development of green manure demonstrative region [39].” (L342-L351) to explain the implementation of green manure policy in Guangxi province.  

4. Some limitations of the research method would be usefull to any readers, so I recommend adding them to the Conclusion section. 

Response: A new paragraph about our research limitations and future work was added in last of this paper: â€śMeanwhile, we observed some limitations of this research to note. On one hand, the estimated perceptions of farmers were not enough robust for application in other provinces of China because the survey was carried out in one province of China. Guangxi province is in the Southern China, and its circumstance is much different with other regions. Therefore, more surveys can be conducted in different provinces to strengthen the reliability and applicability of the findings. On the other hand, this study emphasized that subsidy is one of the key factors that influences the farmers’ willingness of planting green manure, but this just the beginning. Furthermore, the expected value of subsidy should be estimated through survey. Also, apart from cost, others factors, such as the diseases occurred by planting green manure, should be taken into consideration.”  (L372-L380)

Reviewer 2 Report

The paper has good quality and interesting subject. It is aimed to investigate the influence of instrumental variables on farmers’ perception towards sustainable agriculture practices, especially the use of rotation fallow. However, I have recommendations to improve the quality of the paper. Please see my next comments.

The abstract needs further improvements; the authors should add one or two sentences to indicate a introduction of their work

The global novelty of research is not highlighted in the Introduction section as well. The authors should add a paragraph and mention the novelty of this research.

The authors should indicate how this study (results) is going to be beneficial to the policy makers in one paragraph.

Some limitations of the research method would be usefull to any readers, so I recommend adding them to the Conclusion section.

Author Response

Response to Reviewer#2 comments:

Many thanks for your valuable comments and suggestions. We have restructured the discussion section.

1. Generally, the introduction is not well organized and the discussion need to be improved and focused on the data obtained from this study. Response: We have revised the introduction section. We have added several references for some important statements to explain the importance of this study, knowledge gaps and study goals.

Firstly, we have added the sentence â€śIn China, the past half century has seen remarkable growth in food production. In 2016, about 621.4 million tons of grain was produced. Food security is no longer the sole target of agricultural development, and increased attention is now focused on environmental protection and sustainable agricultural development. At present, soil degradation in China affected more than 466 million hectares. It is one of the most serious agricultural, environmental [1] and social-economic problems [2]. Some of those problems were caused by the overuse of chemical fertilizer. The use of chemical fertilizer was passed from 8.8 million tons in 1978 to 58.6 million tons in 2017, either 6.7 times [3]. This excessive utilization generated many environmental problems such as greenhouse gases’ emissions, water pollution, and weak quality of agricultural production and loss of biodiversity [4]. More than 40% of arable land were affected by soil degradation (loss of soil nutrient, salinization, acidification and weakened of ecosystems services) [5]. The implications of that phenomena were food insecurity, negative effects on environment, loss of ecosystems services[1]. Therefore, there are a certain necessity to plant green manure and use rotation fallow to improve soil quality and to protect environment. ” (L29-41) in the first paragraph of the introduction section to explain reality problems of sustainable agricultural development.

 Secondly, we have added the sentence “Although planting green manure in fallow croplands in winter can bring various economic and environmental benefits, including food production, carbon sequestration, soil retention, sandstorm prevention, water retention and provision of habitat for biodiversity. However, implementation of planting green manure practices has been slow, despite their official promotion, and the increasing production cost caused by planting green manure reduces farmers’ enthusiasm to adopt this approach, which is adverse to the sustainability of planting green manure. The cost of planting green manure (Astragalus sinicus L) in paddy fields in south China was 1730 RMB/hm2 [12]. Currently, subsidy policies of planting green manure are still in the research and exploration stage in China. There is no such a subsidy policy for planting green manure nationwide. Under the current policy, the increased production cost caused by planting green manure is fully burdened by farmers, which reduces their willingness to adopt this approach. The cost of green manure planting is high comparatively to economic outcomes and there is no formal or adequate incentive measures which could motivate farmers to planting green manure voluntarily. Even though many researchers recognized that there is a growing environmental awareness and public demand of improvement of environment in China [13]. However, the existing researches paid not much attention to this important topic. A systematic and rigorous study of intention survey of farmers’ willingness for planting green manure has been lacking. What are the key factors that influence farmers’ willingness for planting green manure? How to encourage farmers to more sustainably planting green manure? These questions are very important, but they have not yet attracted enough attentions of scholars.” (L56-L74) in the third paragraph of the introduction section to explain the problems of planting green manure.  

Finally, we have added the sentence “Hence, we carry out the research to estimate farmers’ willingness for planting green manure in typical green manure production regions of China and to identify determinants underlying farmers’ willingness by using simultaneous equation models. The overall goal is to know farmers’ perception towards planting green manure as sustainable agricultural practices and determine which factors affect their willingness for planting green manure. We identify indicators on which efforts could be concentrated so as to sustain a harmonious restoring and develop green manure planting in a specific area. This study is helpful in encouraging farmers more sustainable planting green manure and promoting sustainable development of agriculture.” (L75-L82) in the fourth paragraph of the introduction section to explain the novelty of this research. 

2. Besides, the information provided by the manuscript lacks of substantial novelty. 

Response: We have restructured the discussion section. Firstly, we have added the paragraph â€śIncentive economic policies should be taken to reduce the low economic impact closely associated with green manure planting, specifically, a rotation fallow land in which different practices of agriculture are integrated based on rational use of natural resources, to increase the use of rotation fallow land as sustainable agriculture practice. Therefore, the adoption of rotation fallow allows ecosystems to be maintained sustainably [20, 37]. According to our survey, on average, 38.99% of farmers have participated into planting green manure program and 11.28% of farmers have the interest to use rotation fallow. This result showed that more than 50% of farmers was reluctant to plant green manure and to use rotation fallow as sustainable agriculture practice. This occurs in a context where most farmers were accustomed to overuse chemical fertilizer to maximize yields. This policy has accelerated soil degradation, failure to rotate wet and dry crops and the decrease of planting vegetables and green manure crops [19] and caused enormous consequences to the environment. The modification of traditional agricultural practices could make help a household by finding any opportunities to increase family income and with enough degree to substantially alleviate poverty [38]. This weakness of interest of farmers to adopt planting green manure and to use rotation fallow was in contradiction with the goal of Government of China of raising rural farmers’ income and to protect environment by planting green manure and using rotation fallow as sustainable agriculture practices. ” (L283-L298) to explain problems of low perceptions on planting green manure. 

Secondly, we have added three paragraphs to discuss the result of our research. (L299-334)“According to our research, the first model (M1) showed that farmers’ perception on the use of rotation fallow has a significantly negative coefficient resulting from a combination of seven (7) instrumental variables. The tests of goodness of fit are significant except the test of endogenous. The model has the problem of endogenous. The improvement of consistency of estimators’ coefficients of the second model (M2) required different combinations of two or three instrumental variables. These findings showed that among a big group of many elements, drivers factors are constituted by a small group of elements. Our study found that three instrumental variables are primarily essential to increase the farmers perception on using rotation fallow, and thus high impacted significantly on farmers participation in planting green manure program in Guangxi province because they increase the level of farmers’ perception. The tests of goodness -of- fit are more significant. 

In second model (M2), farmers perception (per) on using rotation fallow had high significantly positive influence on planting green manure while this factor (per) had been influenced by a more significantly positive coefficient of SUPPOL and positive coefficients of SUBFUN and SUBSTAREC . Then, we have the consistency coefficients of estimators in second model (M2). This high consistency of coefficient of estimation is resulting to the reduction of unobserved factors (IV) one by one with the objective of finding the drivers factors of farmers’ perception on using rotation fallow as sustainable agricultural practices. Thus, farmers’ perception impacted more positively and significantly on farmers’ participation in planting green manure. Farmers’ participation in planting green manure will increase by 1.617 times with respect to farmers’ perception on using rotation fallow.

According to our study, especially the driver factor (SUPPOL), farmers’ perception will be greater than the current estimate in the future when considering farmers' preference for kinds of subsidy. This factor is the most suitable query for understanding farmers’ perception for maximizing planting green manure in the future. According to drivers factors, farmers perception is positively affected by standard subsidy of local government of Guangxi based on the area planted (SUBSTAREC) and the farmers’ belief on standard subsidy effects on promoting planting green manure (SUBFUN). Based on survey, 11.31% of respondents had participated in training, 34.23% of respondents had capacity to choose one of three types of subsidy (funds, seeds and mechanical service tools). The farming experience is more significant and had a negative impact on using rotation fallow as sustainable agriculture practice because of 66.67% of respondents have from 0 to 15 years of farming experience. The increase of the training on the green manure planting technology or the demonstration preaching should increase the knowledge of farmers about the importance of green manure planting. By increasing the number of the green manure demonstration basis in different sites in Guangxi province could be a suitable way to enhance the farmer's knowledge in green manure planting technology for agriculture sustainable development. For that, a reasonable standard subsidy for using rotation fallow are needed to reduce the cost of green manure planting activities because green manure planting cost is one of issues of farmers related to expenditure on labor, seeds, weeds management, pest and disease controls, etc.”  

Finally, we have added two paragraphs to indicate how this study is going to be beneficial to the policy makers and to explain the implementation of green manure policy in Guangxi province.  (L335-351) “Clearly, research evidence has shown a reluctance among farmers to participate in planting green manure and in the use of rotation fallow. Therefore, policymakers should urgently focus on strengthen  training sessions and mass awareness on the ecosystem values of green manure and on added-values resulting from rotation fallow practices. The use of Internet, television, radio and others mass media available could serve widely in advertising. Policymakers can also take into account an ecological compensation system based on the use of rotation fallow reinforced by technical guides, technical-financial support and technology to increase the yield and welfare of farmers. To achieve the goal of Chinese government, the environmental and economic incentive policy of technical-financial support for adopting the use of rotation fallow and planting green manure can be a great help to enhance the use of rotation fallow, planting green manure and raising farmers’ income simultaneously. The agriculture’s department of Guangxi has announced a series of guidelines on the development of green manure for the promotion of beautiful villages construction, winter fields, and spring opportunities to develop the agriculture of the leisure and the local tourism because at the same time green manure panting was considered as an important part of the action plan for the increase in grain of 3 million [4]. In 2016, the planting area of green manure is 330,000 hectare and the government provide 15 million Yuan for subsiding seeds of green manure and the development of green manure demonstrative region [39].” 

3. This paper needs to be edited to improve the English. The language using in the current version is difficult to understand the meaning of several sentences. 

Response: We read our paper carefully and modify English language problems, especially follow your instructions of suggestions, and all the modification are in red font.  

4. I sincerely suggest to the authors that, prior to resubmit their manuscript, they should carefully revise and improve it. Major editorial comments are listed below. 

Response: We followed your editorial comments and modified all the mistakes, and revise the references with the correct format. 

Round 2

Reviewer 1 Report

Comments and Suggestions:

    After revised, the quality of the manuscript got improved. The part of introduction provided sufficient background than before.  Discussion paid more attention on the result obtained from this research. The manuscript can be accepted in present form.

Author Response

Dear reviewer,

There is not new comment. We have finished the English editing and upload the new file.

Thank you!

Reviewer 2 Report

The presented paper has been improved after revisions. 

Author Response

(The authors gave the same response as above.)
